# LLEMMA: AN OPEN LANGUAGE MODEL FOR MATHEMATICS

**Zhangir Azerbayev**[1,2]    **Hailey Schoelkopf**[2]    **Keiran Paster**[3,4]

**Marco Dos Santos**[5]    **Stephen McAleer**[6]    **Albert Q. Jiang**[5]    **Jia Deng**[1]

**Stella Biderman**[2]    **Sean Welleck**[6,7]

[1] Princeton University    [2] EleutherAI    [3] University of Toronto    [4] Vector Institute
[5] University of Cambridge    [6] Carnegie Mellon University    [7] University of Washington

## ABSTRACT

We present LLEMMA, a large language model for mathematics. We continue pretraining Code Llama on Proof-Pile-2, a mixture of scientific papers, web data containing mathematics, and mathematical code, yielding LLEMMA. On the MATH benchmark LLEMMA outperforms all known open base models, as well as the unreleased Minerva model suite on an equi-parameter basis. Moreover, LLEMMA is capable of tool use and formal theorem proving without any further finetuning. We openly release all artifacts, including 7 billion and 34 billion parameter models, the Proof-Pile-2, and code to replicate our experiments.[1]

## 1    INTRODUCTION

Language models trained on diverse mixtures of text display remarkably general language understanding and generation capabilities (Brown et al., 2020; Chowdhery et al., 2022), serving as base models that are adapted to a wide range of applications (Raffel et al., 2023). Applications such as open-ended dialogue (Thoppilan et al., 2022; Touvron et al., 2023) or instruction following (Ouyang et al., 2022; Wei et al., 2022) require balanced performance across the entire distribution of natural text, thus favoring *generalist models*. However, if we seek to maximize performance within one domain, such as medicine (Singhal et al., 2022; 2023), finance (Wu et al., 2023), or science (Taylor et al., 2022), a *domain-specific language model* may offer superior capabilities for a given computational cost, or lower computational cost for a given level of capability.

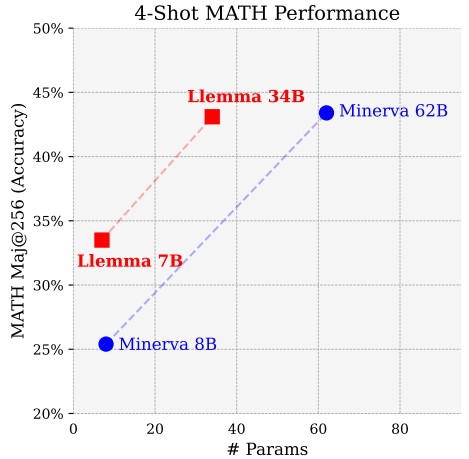

Figure 1: Continued pretraining on Proof-Pile-2 yields LLEMMA, a base model with improved mathematical capabilities.

In this work, we train a domain-specific language model for mathematics. We have several motivations for doing so. First, solving mathematical problems requires pattern matching against a large body of specialized prior knowledge, thus serving as an ideal setting for domain adaptation. Second, mathematical reasoning is in itself a central AI task, its study dating back to at least Gelernter (1959) and Wang (1960) and continuing to today (Lu et al., 2023). Third, language models capable of strong mathematical reasoning are upstream of a number of research topics, such as reward modeling (Uesato et al., 2022; Lightman et al., 2023), reinforcement learning for reasoning (Polu et al., 2022; Lample et al., 2022), and algorithmic reasoning (Zhou et al., 2022; Zhang et al., 2023).

---

[1]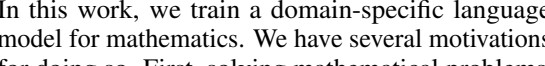 https://github.com/EleutherAI/math-lm

Although domain-specific models for mathematics have been trained in the past, they have either been closed access (Lewkowycz et al., 2022), limiting their ability to become a platform for further research, or have lagged far behind the closed access state-of-the-art (Azerbayev et al., 2023).

We present a recipe for adapting a language model to mathematics through continued pretraining (Lewkowycz et al., 2022; Rozière et al., 2023) on Proof-Pile-2, a diverse mixture of math-related text and code. Applying the recipe to Code Llama (Rozière et al., 2023) yields LLEMMA: 7 billion and 34 billion parameter base language models with substantially improved mathematical capabilities.

Specifically, our contributions are as follows:

1. We train and release the LLEMMA models: 7B and 34B parameter language models specialized for mathematics. The LLEMMA models are a new state-of-the-art for publicly released base models on MATH (Lewkowycz et al., 2022).

2. We release the AlgebraicStack, a dataset of 11B tokens of code specifically related to mathematics.

3. We demonstrate that LLEMMA is capable of using computational tools to solve mathematical problems, namely, the Python interpreter and formal theorem provers.

4. Unlike prior mathematics language models such as Minerva (Lewkowycz et al., 2022), the LLEMMA models are open access and we open source our training data and code. This allows LLEMMA to serve as a platform for future research in mathematical reasoning.

Our work builds on findings in Minerva (Lewkowycz et al., 2022), but differs in several ways: (1) LLEMMA's training and evaluation covers a wider range of data and tasks, notably code data (e.g., the AlgebraicStack), tool use, and formal mathematics; (2) our work only depends on publicly accessible tools and data; (3) we provide new analyses related to the continued training data mixture, memorization, and additional supervised finetuning; (4) we make all artifacts publicly available.

## 2 APPROACH

LLEMMA models are 7 billion and 34 billion parameter language models specialized for mathematics. Our approach is to continue pretraining Code Llama (Rozière et al., 2023) on the Proof-Pile-2.

| Model | Adaptation tokens | Open |
|---|---|---|
| Minerva-8b | 164B | ✗ |
| Minerva-62b | 109B | ✗ |
| LLEMMA-7b (ours) | 200B | ✓ |
| LLEMMA-34b (ours) | 50B | ✓ |

| Dataset | Tokens | Open |
|---|---|---|
| Minerva Dataset | 38.5B | ✗ |
| Proof-Pile-2 (ours) | 55B | ✓ |
|   Code (AlgebraicStack) | 11B | ✓ |
|   OpenWebMath (Paster et al., 2023)) | 15B | ✓ |
|   ArXiv (Computer, 2023)) | 29B | ✓ |

Figure 2: Comparison of LLEMMA and Minerva training

### 2.1 DATA: Proof-Pile-2

We form the Proof-Pile-2, a 55B-token mixture of scientific papers, web data containing mathematics, and mathematical code. With the exception of the Lean proofsteps subset (see Appendix B), the Proof-Pile-2 has a knowledge cutoff of April 2023.

**Code.** Computational tools such as numerical simulations, computer algebra systems, and formal theorem provers are of ever increasing importance to mathematicians (Avigad, 2018). Motivated by this fact, we create AlgebraicStack, an 11B-token dataset of source code from 17 languages, spanning numerical, symbolic, and formal math. The dataset consists of filtered code from the Stack (Kocetkov et al., 2022), public GitHub repositories, and formal proofstep data. Table 9 shows the number of tokens by language in AlgebraicStack. See Appendix B.1 for further details on AlgebraicStack.

**Web data.** We use OpenWebMath (Paster et al., 2023), a 15B-token dataset of high-quality web pages filtered for mathematical content. OpenWebMath filters CommonCrawl web pages based

on math-related keywords and a classifier-based math score, preserves mathematical formatting (e.g., LaTeX, AsciiMath), and includes additional quality filters (e.g., perplexity, domain, length) and near-deduplication. Refer to Paster et al. (2023) for a full description of OpenWebMath.

**Scientific papers.** We use the ArXiv subset of RedPajama (Computer, 2023), an open-access reproduction of the LLaMA training dataset. The ArXiv subset contains 29B tokens.

**General natural language and code data.** Following Lewkowycz et al. (2022), our training mixture consists of a small amount of general domain data, which functions as a form of regularization. Since the pretraining dataset for LLaMA 2 is undisclosed, we use the Pile (Gao et al., 2020; Biderman et al., 2022) as a surrogate training dataset. We set 95% of our training mixture to be the Proof-Pile-2, 2% to be from the Pile (with ArXiv removed, as it is separately in Proof-Pile-2), and 3% to be the GitHub subset of RedPajama (Computer, 2023).

Further information on dataset composition and a datasheet are in Appendix B and Appendix E, respectively. We publicly release Proof-Pile-2 at `hf.co/datasets/EleutherAI/proof-pile-2`.

## 2.2 MODEL AND TRAINING

Each model is initialized from Code Llama (Rozière et al., 2023). Code Llama models are decoder-only transformer language models initialized from Llama 2 (Touvron et al., 2023) and further trained on 500B tokens of code. We continue training the Code Llama models on Proof-Pile-2 using a standard autoregressive language modeling objective. We train the 7B model for 200B tokens, and the 34B model for 50B tokens.

We train all models in bfloat16 mixed precision using the GPT-NeoX library (Andonian et al., 2023) across 256 A100 40GB GPUs. We use Tensor Parallelism (Shoeybi et al., 2019) with a world size of 2 for LLEMMA-7B , and a world size of 8 for LLEMMA-34B, alongside ZeRO Stage 1 sharded optimizer states (Rajbhandari et al., 2020) across Data Parallel (Goyal et al., 2017) replicas. We use Flash Attention 2 (Dao, 2023) to improve throughput and further reduce memory requirements.

LLEMMA 7B is trained for $42,000$ steps with a global batch size of 4 million tokens and a 4096 token context length. This corresponds to roughly $23,000$ A100-hours. The learning rate is warmed up to $1 \cdot 10^{-4}$ over 500 steps, then set to cosine decay to $1/30$th of the maximum learning rate over $48,000$ steps. The reason for the discrepancy between the number of training steps and the scheduler length is that we planned to train for $48,000$ steps, but encountered NaN losses after step $42,000$, likely caused by unstable optimization or hardware failures (Elsen et al., 2023).

LLEMMA 34B is trained for $12,000$ steps with a global batch size of 4 million tokens and a 4096 context length. This corresponds to roughly $47,000$ A100-hours. The learning rate is warmed up to $5 \cdot 10^{-5}$ over 500 steps, then decayed to $1/30$th the peak learning rate.

Before training LLEMMA 7B, we contract the RoPE (Su et al., 2022) base period of the Code Llama 7B initialization from $\theta = 1,000,000$ to $\theta = 10,000$. This is so that the long context finetuning procedure described in Peng et al. (2023)and Rozière et al. (2023) can be repeated on the trained LLEMMA 7B (we leave actually doing so to future work). Due to compute constraints, we were unable to verify that training LLEMMA 34B with a contracted RoPE base period did not come with a performance penalty, therefore for that model we preserved $\theta = 1,000,000$.

## 3 EVALUATION

Our goal is to evaluate LLEMMA as a base model for mathematical text. To this end, we compare LLEMMA models using few-shot evaluation (Brown et al., 2020), and primarily focus on state-of-the-art models that have not been finetuned on supervised examples for the task. First, we evaluate the model's ability to solve mathematics problems using chain of thought reasoning (Wei et al., 2023) and majority voting (Wang et al., 2023). Our evaluations include MATH (Hendrycks et al., 2021b) and GSM8k (Cobbe et al., 2021), the de-facto standard benchmarks for evaluating quantitative reasoning in language models (Lewkowycz et al., 2022). Second, we explore few-shot tool use and formal theorem proving. Third, we study the effects of memorization and the data mixture. Appendix G contains a preliminary study of supervised finetuning with LLEMMA.

### 3.1 Chain-of-thought mathematical problem solving

These tasks involve generating self-contained text solutions to problems expressed in LaTeX or natural language, without using external tools (Lewkowycz et al., 2022). We use the following evaluation:

- **MATH** (Hendrycks et al., 2021b), a dataset with 12.5k problems (5k evaluation) from high-school math competitions. Given a problem statement, the model generates a LaTeX solution and an answer that must match a reference answer. We follow a similar task implementation to Lewkowycz et al. (2022), using their four-example prompt and evaluating answers for exact string match or `SymPy` equivalence.

- **GSM8k** (Cobbe et al., 2021), a dataset of middle-school level math word problems. We use the 8-shot prompt from Wei et al. (2023), as Lewkowycz et al. (2022) do not specify their evaluation prompt or number of few-shot examples.

- **OCWCourses** (Lewkowycz et al., 2022), a collection of undergraduate-level STEM problems harvested from MIT's OpenCourseWare. We use the four-example prompt provided by (Lewkowycz et al., 2022).

- **MMLU-STEM** (Hendrycks et al., 2021a), a subset of 18 out of 57 subjects in the MMLU benchmark. We follow Lewkowycz et al. (2022) and use their provided four-example chain-of-thought prompt.

- **SAT**, we create a dataset consisting of the 32 math questions that do not contain figures from the May 2023 College Board SAT examination, which is after our model's knowledge cutoff.

---

**Input:**
Let $f(r) = \sum_{j=2}^{2008} \frac{1}{j^r} = \frac{1}{2^r} + \frac{1}{3^r} + \cdots + \frac{1}{2008^r}$. Find $\sum_{k=2}^{\infty} f(k)$.

**LLEMMA 34B solution:**
We have

$$\sum_{k=2}^{\infty} f(k) = \sum_{k=2}^{\infty} \sum_{j=2}^{2008} \frac{1}{j^k} = \sum_{j=2}^{2008} \sum_{k=2}^{\infty} \frac{1}{j^k} = \sum_{j=2}^{2008} \frac{1}{j^2} \sum_{k=0}^{\infty} \frac{1}{j^k} = \sum_{j=2}^{2008} \frac{1}{j^2} \frac{1}{1 - \frac{1}{j}}$$

$$= \sum_{j=2}^{2008} \frac{1}{j(j-1)} = \sum_{j=2}^{2008} \left( \frac{1}{j-1} - \frac{1}{j} \right)$$

$$= \left( \frac{1}{1} - \frac{1}{2} \right) + \left( \frac{1}{2} - \frac{1}{3} \right) + \cdots + \left( \frac{1}{2007} - \frac{1}{2008} \right)$$

$$= 1 - \frac{1}{2008}$$

$$= \boxed{\frac{2007}{2008}}.$$

Final Answer: The final answer is $\frac{2007}{2008}$.

---

Figure 3: Example of a LLEMMA 34B solution to a MATH (Hendrycks et al., 2021a) problem. This problem is tagged with difficulty level 5, the highest in MATH. The model was conditioned on the 4-shot prompt described in subsection 3.1, and the solution was produced by greedy decoding. The model had to apply two nontrivial steps to solve this problem: (1) noticing that swapping the order of summation simplifies the problem, and (2) noticing that the resulting sum telescopes.

We compare with Minerva (Lewkowycz et al., 2022), which continued pretraining the PaLM language model on a dataset of technical content; Code Llama, the initialization of LLEMMA's continued pretraining; and Llama 2, the initialization of Code Llama's continued pretraining on code. For open access models, we report scores computed using our evaluation suite, which is implemented as a fork of the Language Model Evaluation Harness (Gao et al., 2021). For Minerva models, we report benchmark scores from Lewkowycz et al. (2022).

**Results.** LLEMMA's continued pretraining on Proof-Pile-2 improves few-shot performance on the five mathematical benchmarks. LLEMMA 34B improves over Code Llama by 20 percentage points on GSM8k and 13 points on MATH, and LLEMMA 7B outperforms the proprietary Minerva model. Our approach also outperforms all open-weight language models at the time of writing. We conclude that continued pretraining on Proof-Pile-2 is effective for improving a pretrained model's ability to perform mathematical problem solving.

LLEMMA is pretrained on a diverse distribution of mathematics-related data, and is not tuned for a particular task. Therefore, we expect that LLEMMA can adapt to many other tasks via task-specific finetuning and few-shot prompting.

|  |  | GSM8k | OCW | MMLU-STEM | SAT | MATH |
|---|---|---|---|---|---|---|
| Llama 2 | 7B | 11.8% | 3.7% | 29.9% | 25.0% | 3.2% |
| Code Llama | 7B | 10.5% | 4.4% | 25.1% | 9.4% | 4.5% |
| Minerva | 8B | 16.2% | 7.7% | 35.6% | - | 14.1% |
| LLEMMA | 7B | **36.4%** | **7.7%** | **37.7%** | **53.1%** | **18.0%** |
| Code Llama | 34B | 29.6% | 7.0% | 40.5% | 40.6% | 12.2% |
| LLEMMA | 34B | **51.5%** | **11.8%** | **49.0%** | **71.9%** | **25.0%** |
| Minerva | 62B | 52.4% | 12.0% | 53.9% | - | 27.6% |
| Minerva | 540B | 58.8% | 17.6% | 63.9% | - | 33.6% |

Table 1: Results on our five chain-of-thought reasoning tasks with samples generated via greedy decoding. Minerva results are quoted from Lewkowycz et al. (2022). Note that CodeLlama 7B performs worse than random guessing (25%) on MMLU and SAT, largely due to failing to conclude its chain of thought with a valid answer.

|  |  | GSM8k maj@$k$ | OCW maj@$k$ | MMLU-STEM maj@$k$ | SAT maj@$k$ | MATH maj@$k$ |
|---|---|---|---|---|---|---|
| Minerva | 8B | 28.4% | 12.5% | 43.4% | - | 25.4% |
| LLEMMA | 7B | **54.0%** | **14.3%** | **49.9%** | **78.1%** | **33.5%** |
| LLEMMA | 34B | **69.3%** | **18.4%** | **59.7%** | **81.3%** | **43.1%** |
| Minerva | 62B | 68.5% | 23.5% | 63.5% | - | 43.4% |
| Minerva | 540B | 78.5% | 30.8% | 75.0% | - | 50.3% |

Table 2: Majority voting results for LLEMMA and Minerva. Minerva results are quoted from Lewkowycz et al. (2022). Voting is done with $k = 256$ for MATH, $k = 100$ for GSM8k and OCW, and $k = 16$ for MMLU-STEM and SAT. We sample with temperature $T = 0.6$ for $k = 256$ and $k = 100$ and $T = 0.3$ for $k = 16$, and use nucleus sampling with $p = 0.95$ (Holtzman et al., 2020). Due to compute constraints, we do not calculate majority voting scores for Llama 2 and Code Llama.

## 3.2 MATHEMATICAL PROBLEM SOLVING WITH TOOL USE

These tasks involve solving problems with access to computational tools. We evaluate the following:

- **MATH+Python**, the model is prompted to alternately describe a solution step in natural language, then execute that step with code. The final answer is a program that executes to a numeric type or a `SymPy` object. Our few-shot prompt includes examples that use built-in numeric operations, the `math` module, and `SymPy`.

- **GSM8k+Python**, solving a GSM8k word problem by writing a Python program that executes to an integer answer. We use the prompt from Gao et al. (2023).

**Results.** As seen in Table 3, LLEMMA improves over Code Llama on both tasks. Its performance on MATH and GSM8k with tools is also higher than its performance on these datasets without tools.

|  |  | GSM8k+Python pass@1 | MATH+Python pass@1 |
|---|---|---|---|
| Code Llama | 7B | 27.1% | 17.2% |
| LLEMMA | 7B | 40.1% | 21.5% |
| Code Llama | 34B | 52.7% | 23.5% |
| LLEMMA | 34B | 62.6% | 27.1% |

Table 3: Mathematical problem solving with tool use.

## 3.3 FORMAL MATHEMATICS

Interactive proof assistants such as Lean (de Moura et al., 2015), Isabelle (Wenzel et al., 2008), and Coq (Paulin-Mohring, 1989a;b) express mathematics in programming languages that allow for verification. These languages are data scarce compared to mainstream languages, especially in the context of pretraining. For instance, the Stack dataset used to pretrain language models in the BigCode project (Allal et al., 2023) has over 700 gigabytes of Python, compared to 322 megabytes of Lean. Proof assistants also require models to leverage information that is not present in raw source code, such as goal states that contain information about each step of a proof.

---

**Problem (MATH Number theory 185):** When a number is divided by 5, the remainder is 3. What is the remainder when twice the number is divided by 5? Show that it is 1.

- - - - - - - - - - - - - - - - - - - - - - - - - - - - - - - - - - - - - - - - - - - - - - - - - - -

**Human-written informal proof:** If our number is $n$, then $n \equiv 3 \pmod 5$. This tells us that

$$2n = n + n \equiv 3 + 3 \equiv 1 \pmod 5.$$

The remainder is 1 when the number is divided by 5.

---

**Informal-to-formal (Isabelle):**
{Problem, human-written informal proof}

```
theorem mathd_numbertheory_185:
  fixes n ::nat
  assumes "n mod 5 = 3"
  shows "(2 * n) mod 5 = 1"
```
- - - - - - - - - - - - - - - - - - - - - - - - - - -
```
proof -
  have "2 * n = n + n"  <ATP>
  also have "... mod 5 =
    (n mod 5 + n mod 5) mod 5"  <ATP>
  also have "... = (3 + 3) mod 5"
    using assms <ATP>
  also have "... = 1"  <ATP>
  finally show ?thesis <ATP>
qed
```

**Formal-to-formal (Lean 4):**

```
theorem mathd_numbertheory_185
  (n : ℕ) (h₀ : n % 5 = 3)
  : 2 * n % 5 = 1 := by

-- INPUT (step 1):
--   n: ℕ
--   h₀: n % 5 = 3
--   ⊢ 2 * n % 5 = 1
rw [mul_mod, h₀]

-- INPUT (step 2):
--   n: ℕ
--   h₀: n % 5 = 3
--   ⊢ 2 % 5 * 3 % 5 = 1
simp only [h₀, mul_one]
```

Figure 4: Example formal proofs from LLEMMA-7b. *Left:* The model is given a problem, informal proof, and formal statement, following Jiang et al. (2023). It generates a formal proof (starting with `proof -`) containing Isabelle code and calls to automation (shown as *<ATP>*). *Right:* The model is given a proof state, visualized as a grey comment, and generates the subsequent step (e.g. `rw [..`).

Proof-Pile-2's AlgebraicStack contains over 1.5 billion tokens of formal mathematics data, including proof states extracted from Lean and Isabelle formalizations. While a full investigation of formal math is outside the scope of this paper, we evaluate LLEMMA few-shot on two tasks:

- **Informal-to-formal proving** (Jiang et al., 2023), the task of generating a formal proof, given a formal statement, an informal LaTeX statement, and an informal LaTeX proof. The formal proof is checked by the proof assistant. We use the Isabelle proof assistant and evaluate on miniF2F (Zheng et al., 2021), a benchmark consisting of problem statements from Olympiads and undergraduate coursework. For the prompt, we use 11 (formal statement, informal statement, informal proof, formal proof) examples from Jiang et al. (2023), selecting 7 examples for number theory problems, and 6 examples for all others. We generate a single proof with greedy decoding.

- **Formal-to-formal proving** (e.g., Polu & Sutskever (2020)), the task of proving a formal statement by generating a sequence of proof steps (tactics). At each step, the input is a state $x_t$ given by the proof assistant, and the language model's task is to generate a proof step $y_t$ (a sequence of code). The proof step is checked by the proof assistant, yielding a new state $x_{t+1}$ or an error message. The process continues, stopping if a proof is completed or a timeout is reached. We prompt the model using three $(x_t, y_t)$ examples. We evaluate on miniF2F (Zheng et al., 2021) using the Lean 4 proof assistant, and use a standard best first search. See Appendix D for more details.

**Results.** As seen in Table 4, LLEMMA's continued pretraining on Proof-Pile-2 improved few-shot performance on the two formal theorem proving tasks.

| Method | Informal-to-formal | | | Method | Formal-to-formal | |
|---|---|---|---|---|---|---|
| | miniF2F-valid | miniF2F-test | | | Search | miniF2F-test |
| Sledgehammer | 14.72% | 20.49% | | ReProver (fine-tuned) | 1×64 | 26.50% |
| Code Llama 7b | 16.31% | 17.62% | | Code Llama 7b | 1×32 | 20.49% |
| Code Llama 34b | 18.45% | 18.03% | | Code Llama 34b | 1×32 | 22.13% |
| | | | | COPRA (GPT-4) | -[†] | 23.36% |
| LLEMMA-7b | 20.60% | 22.13% | | LLEMMA-7b | 1×32 | 26.23% |
| LLEMMA-34b | 21.03% | 21.31% | | LLEMMA-34b | 1×32 | 25.82% |

Table 4: Formal theorem proving tasks. *Left*: Informal-to-formal proving in Isabelle, showing the percentage of proven theorems with greedy decoding. *Right*: Formal-to-formal proving in Lean, showing the percentage of proven theorems with the given number of attempts × generations-per-iteration of best first search, and a 10-minute timeout. Sledgehammer (Paulson & Nipkow, 2023) is built-in Isabelle automation. ReProver (Yang et al., 2023) is a supervised and retrieval-augmented model. COPRA (Thakur et al., 2023) is a retrieval-augmented GPT-4 based method. [†] COPRA does not use best first search, but instead samples from GPT-4 (OpenAI, 2023) a maximum of 60 times.

On informal-to-formal proving, LLEMMA-7b closes 22.1% of the theorems, improving upon its Code Llama initialization and the Sledgehammer prover. The theorems that LLEMMA proves are often complementary to those proved with Sledgehammer: taking the union of Sledgehammer and LLEMMA proofs results in 26 new validation proofs (an 11 percentage-point increase), and 17 new test proofs (a 7 point increase); see Appendix Table 11. Prior to our work, the only demonstration of few-shot proof autoformalization used the proprietary Codex model (Jiang et al., 2023).

On Lean 4 formal-to-formal proving, LLEMMA-7b improves upon its Code Llama initialization, and performs similar to ReProver (Yang et al., 2023), a retrieval-augmented language model finetuned for tactic prediction. LLEMMA adapts to the task using a 3 example prompt, which to our knowledge is the first demonstration of few-shot tactic prediction for theorem proving by an open model.

## 3.4 IMPACT OF DATA MIXTURE

When training a language model, it is common to upsample high-quality subsets of the training data according to mixture weights (Brown et al., 2020; Gao et al., 2020; Xie et al., 2023). We select mixture weights by doing short training runs on several hand-picked mixture weights, then choosing the one which minimizes perplexity on a set of high-quality held-out text (we use the MATH training set). Table 5 shows the MATH training set perplexity of models trained using different mixtures of arXiv to web to code. Based on these results, we trained LLEMMA with a ratio of $2:4:1$. Note that our methodology uses the MATH training set to determine a training hyperparameter, though we expect that the effect is similar to that of related high-quality texts.

| Mixture | MATH training set perplexity | | | | | | | |
|---|---|---|---|---|---|---|---|---|
| | Overall | Prealgebra | Algebra | Number Theory | Counting & Probability | Geometry | Intermediate Algebra | Precalculus |
| **2:4:1** | **1.478** | **1.495** | **1.515** | **1.552** | **1.475** | **1.519** | **1.439** | **1.331** |
| 2:4:2 | 1.482 | 1.500 | 1.519 | 1.556 | 1.477 | 1.524 | 1.443 | 1.334 |
| 4:2:1 | 1.487 | 1.505 | 1.524 | 1.561 | 1.481 | 1.534 | 1.447 | 1.338 |
| 4:2:2 | 1.489 | 1.508 | 1.527 | 1.562 | 1.483 | 1.538 | 1.447 | 1.339 |
| 4:4:1 | 1.487 | 1.506 | 1.525 | 1.561 | 1.482 | 1.529 | 1.446 | 1.335 |
| 4:4:2 | 1.485 | 1.503 | 1.523 | 1.559 | 1.480 | 1.529 | 1.444 | 1.334 |

Table 5: MATH training set perplexity of Code Llama 7B models trained using different data mixtures for a reduced number of steps. Each mixture is represented by its arXiv:Web:Code ratio.

### 3.5 DATASET OVERLAP AND MEMORIZATION

**Do test problems or solutions appear in the corpus?** We check whether any 30-gram in a test sequence (either an input problem or an output solution) occurs in any OpenWebMath or AlgebraicStack document. If so, we say that a *hit* occurred between the sequence and the document. Table 6 shows hits between sequences from MATH and documents from Proof-Pile-2. Using our methodology, around 7% of MATH test problem statements and 0.6% of MATH test solutions have hits. Note that our methodology gives a lower bound on the number of semantically equivalent sequences (e.g., it does not account for alternative phrasing).

We manually inspected 100 uniformly sampled hits between a test problem statement and an Open-WebMath document. 41 of the cases had no solution, which included websites with a list of problems, discussions, or hints. 49 had an alternative solution to the MATH ground-truth solution, but with the same answer. These include solutions that solve the problem differently than the ground-truth, solutions with missing details, and discussions that include the answer. 9 cases had a missing or incorrect answer, and 1 had the same solution as in the ground-truth. In summary, we find that solutions can appear in a corpus derived from web documents, particularly alternative solutions to those in the evaluation set. We repeated our analysis with 20-gram hits and our findings were similar, though with false positives; see Appendix Figure 6 for examples.

| Proof-Pile-2 | Test | Problem | | Solution | |
|---|---|---|---|---|---|
| | | Example | Docs | Example | Docs |
| OpenWebMath | MATH | 348 | 717 | 34 | 46 |
| AlgebraicStack | MATH | 3 | 3 | 1 | 1 |
| OpenWebMath | GSM8k | 2 | 3 | 0 | 0 |
| AlgebraicStack | GSM8k | 0 | 0 | 0 | 0 |

| | |
|---|---|
| Same solution | 1 |
| Different solution, same answer | 49 |
| Different solution, different answer | 9 |
| No solution | 41 |
| Different problem | 0 |

Table 6: *Left:* 30-gram hits between MATH test problems or solutions and Proof-Pile-2 documents. *Example* and *Docs* are the numbers of unique test examples and Proof-Pile-2 documents with a hit. *Right:* manual inspection of 100 hits between a problem statement and a Proof-Pile-2 document.

**How do problems in the corpus impact performance?**
Next, we evaluate LLEMMA-34b on the test examples with a 30-gram hit, and the test examples without a 30-gram hit. Table 7 shows the accuracy partitioned by MATH difficulty level. The model's accuracy remains low on difficult problems (e.g., 6.08% on Level 5 problems with a hit, versus 6.39% on problems without a hit), and we observe no clear relationship between 30-gram hits and accuracy across difficulty levels. We conclude that a nontrivial match between a test example and a training document did not imply that the model generated a memorized correct answer. We repeated the analysis with 20-grams and with the 7b model, and our findings were analogous. Figure 7 shows an example.

| MATH Level | Hit Accuracy | Nonhit Accuracy | # Hits |
|---|---|---|---|
| Level 1 | 72.73 | 61.50 | 11 |
| Level 2 | 35.71 | 40.18 | 28 |
| Level 3 | 30.36 | 26.88 | 56 |
| Level 4 | 14.89 | 16.61 | 94 |
| Level 5 | 6.08 | 6.39 | 181 |

Table 7: LLEMMA-34b's accuracy on hits (a 30-gram overlap between a problem or solution and a training sequence) and non-hits by MATH difficulty level.

Finally, we check 30-gram hits between LLEMMA's MATH generations and OpenWebMath. There were 13 hits, which occurred when the model generated a common sequence of numbers (e.g., a list of Fibonacci numbers), plus one instance of factoring a polynomial. Appendix Figure 6 shows an example. We find all of these observations worthy of further study. Using LLEMMA and Proof-Pile-2 to better understand data, memorization, and performance is an interesting future direction. We include the code for our analysis in the LLEMMA repository.

## 4 RELATED WORK

**Large-scale language modeling.** Recent progress in large language models involves two connected threads: the increasing scale of models and data (Hoffmann et al., 2022; Kaplan et al., 2020; Chowdhery et al., 2022), and a progression toward more generalist models (Radford et al., 2019; Brown et al., 2020) which are capable of solving diverse problems and adapting quickly to novel tasks. A third thread relates to enabling open access to language models with these capabilities (Black et al., 2022; Biderman et al., 2023; Touvron et al., 2023; Rozière et al., 2023). Our work provides a recipe for specializing these language models to the domain of mathematics, providing a platform for further research and applications.

**Domain adaptation.** Language model applications typically require a general-domain pretraining step, followed by a shorter fine-tuning step. The finetuning step is often aimed at imbuing instruction-following ability (Sanh et al., 2022; Wei et al., 2022) or aligning a model's outputs with human preferences (Ziegler et al., 2019; Ouyang et al., 2022; Bai et al., 2022). Other work explores adapting pretrained models to novel domains by continued training (Rozière et al., 2023; Beltagy et al., 2019), parameter-efficient finetuning methods (Yong et al., 2023), retrieval augmentation (Min et al., 2023; Asai et al., 2023), and other techniques. We provide an adaptation recipe involving continued training and targeted data collection.

**Language models for mathematics.** Applying large language models to problems in mathematics is an active subfield of machine learning, including benchmarking mathematical knowledge and reasoning at varying levels (Hendrycks et al., 2021b; Zheng et al., 2021; Welleck et al., 2022; Azerbayev et al., 2023). Although achieving strong mathematical reasoning is an important target, it is difficult to assess the correctness of models' answers and processes, especially as models become more capable (Bowman et al., 2022; Uesato et al., 2022; Lightman et al., 2023; Cobbe et al., 2021).

A number of recent works focus on supervised finetuning on task-relevant (input, output) pairs (e.g.,Yu et al. (2023); Yue et al. (2023)). Doing so boosts performance on some common mathematical language modeling benchmarks, but trains the model for these specific tasks. In contrast, Lewkowycz et al. (2022) and our work seek to train a *base* language model as a platform for further development.

**Language models for formal mathematics.** An ongoing line of work explores integrating language models with interactive proof assistants in the context of mathematics. This includes synthesizing proofs via tactic prediction (Polu & Sutskever, 2020; Han et al., 2022; Lample et al., 2022; Jiang et al., 2022), autoformalization (Wu et al., 2022; Jiang et al., 2023), and integrated tools (Welleck & Saha, 2023). Due to high computational costs of search, language models applied to this domain have traditionally been small, but recent work has demonstrated promise in the use of larger models (First et al., 2023; Jiang et al., 2023). Our work provides a demonstration of few-shot proof autoformalization and tactic prediction, a large collection of formal mathematics data, along with an open access model for further exploring these directions.

## 5 CONCLUSION

We introduce LLEMMA and Proof-Pile-2, a novel base model and corpus for language modeling of mathematics. Our models, dataset, and code are openly available. We have shown that LLEMMA achieves state-of-the-art results for open-weights models on mathematical problem solving benchmarks, shown capabilities of using external tools via Python code, and demonstrated few-shot tactic prediction for theorem proving. We hope that LLEMMA and Proof-Pile-2 will be a useful base for future work on understanding language model generalization and dataset composition, investigating the limits of domain-specific language models, using language models as tools for mathematicians, and improving the mathematical capabilities of language models.

## ACKNOWLEDGEMENTS

We would like to thank Dragomir Radev, Arman Cohan, Jesse Michael Han, and the Deepmind Blueshift team for valuable guidance. We thank Jonah Philion for the model name. We thank Aviya Skowron for advising us on ethical considerations in the development and release of our models. We thank Jonathan Laurent and Leo Du for contributions to our open-source code.

We would also like to thank several parties for donating computing resources for this project: Stability AI (training the LLEMMA models), CoreWeave (evaluations and finetuning), the Province of Ontario and companies sponsoring the Vector Institute for Artificial Intelligence (`www.vectorinstitute.ai/partners`), and Brigham Young University (finetuning). KP is supported by an NSERC PGS-D award.

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

## A    AUTHOR CONTRIBUTIONS

**Training Data.** Zhangir Azerbayev, Keiran Paster, Marco Dos Santos, Sean Welleck.

**Model training.** Zhangir Azerbayev, Hailey Schoelkopf, Keiran Paster.

**Evaluations.** Zhangir Azerbayev, Hailey Schoelkopf, Keiran Paster, Marco Dos Santos, Stephen McAleer, Albert Q. Jiang, Sean Welleck.

**Formal math evaluations.** Sean Welleck.

**Memorization analysis.** Sean Welleck, Keiran Paster.

**Senior Authorship and Advising.** Jia Deng, Stella Biderman, Sean Welleck.

## B    DATA: Proof-Pile-2

| Data source | Tokens | Weight |
|---|---|---|
| Proof-Pile-2 | 55B | – |
|     Code (AlgebraicStack) | 11B | 1.00 |
|     Web (OpenWebMath) | 15B | 4.00 |
|     Papers (ArXiv) | 29B | 2.00 |
| General code (RedPajama) | 59B | 0.22 |
| General language (Pile) | 300B | 0.15 |

Table 8: Proof-Pile-2 data sources (top), general language and code data included during training (bottom), and the mixture weights of each component during training.

### B.1    MATHEMATICAL CODE: AlgebraicStack

AlgebraicStack contains roughly 11B tokens of code related to mathematics. We describe its sources, filtering, and content below. Table 9 shows the number of tokens per language in AlgebraicStack.

| Language | AlgebraicStack **tokens** | Language | AlgebraicStack **tokens** |
|---|---|---|---|
| Agda | 35.2 M | Julia | 531.0 M |
| C | 25.1 M | Jupyter | 199.1 M |
| C++ | 954.1 M | Lean | 285.6 M |
| Coq | 281.9 M | Maple | 2.0 M |
| Fortran | 724.9 M | Matlab | 65.8 M |
| GAP | 3.6 M | Python | 6,098.8 M |
| Haskell | 9.1 M | R | 71.3 M |
| Idris | 10.9 M | Tex | 567.7 M |
| Isabelle | 1,089.7 M | **Total** | **10,955.7 M** |

Table 9: Tokens in AlgebraicStack, computed with the Llama tokenizer.

#### B.1.1    GITHUB CODE

The following programming languages were either barely present in the Stack or consisted of largely incorrect filetypes, so we downloaded data for these languages directly via the Github Python API.

- **Coq** : We filter for files with the `.v` extension, and include Coq via including files that match a heuristic filter for the keywords **"Theorem"**, **"Proof"**, **"Qed"**, **"Inductive"**, **"Definition"**, **"Fixpoint"** and exclude Verilog files via the keyword blacklist **"pragma"**, **"endmodule"**, **"posedge"**, **"negedge"**, **"wire"**. We additionally exclude files noted as automatically generated.

- **Isabelle** : We filter for files with the `.thy` extension and include files matching the keyword whitelist **"theorem "**, **"lemma "**. We keep only `isabelle-prover/mirror-afp-devel` and discard all other older copies of the Archive of Formal Proofs. We further remove theorem statements and proofs that have a theorem name in the PISA (Jiang et al., 2021) test set.
- **Lean** : We filter for files with the `.lean` extension, using the keyword whitelist **"theorem "**, **"lemma "**, **"example "**. We remove all dependency files, and in order to avoid known benchmark contamination, we blacklist the `ProofNet` and `MiniF2F` repositories. We further remove theorems or lemmas that share a theorem name with the LeanDojo (Yang et al., 2023) val or test sets.
- **MATLAB** : We filter for files with the `.m` extension, using the keyword whitelist **"#import"**, **"interface"**, **"implementation"**, **"property"**, and blacklist C files via the keywords **"#include"** and the regex `r' main\(.*{$'`

We implemented a cutoff date for our Github API downloads, and used a cutoff date of April 1, 2023.

For all languages, unless otherwise stated, we additionally filtered out files with a filesize greater than $1048575$ bytes or with a numerical density (ratio of digit characters to non-digit characters) of $0.5$. We additionally perform document-level exact deduplication by removing documents which contain an overlapping 2048-character chunk as another document.

### B.1.2 LEAN PROOFSTEPS

We extract a dataset of (tactic state, next tactic) pairs from Mathlib 4 (mathlib Community, 2020) using the `lean-training-data` (Morrison, 2023) tool. We use Mathlib 4 commit `c779bd5`, which was created on August 20th 2023.

### B.1.3 ISABELLE PROOFSTEPS

We construct a dataset of Isabelle proofs, building upon the PISA dataset Jiang et al. (2021). Isabelle Proofsteps comprises proofs from the Archive of Formal Proofs and Isabelle Standard Library, scraped with PISA Jiang et al. (2021). Each entry in the dataset includes the theorem statement, the proof states and the proof steps, separated by specific tags. To maintain the integrity of evaluations using the PISA test set, we decontaminate Isabelle Proofsteps by removing theorems whose names overlap with those in the PISA test set. Although this approach results in a strict filtering – removing more than 10,000 theorems although there are only 3600 in the PISA test set – we consider it acceptable in order to mitigate data contamination. After filtering, Isabelle Proofsteps contains 251,000 theorems.

### B.1.4 STACK FILTERING

We source the following programming languages from the Stack (Kocetkov et al., 2022) dataset, and describe our filtering process and quality issues we chose to mitigate beyond our default quality heuristics:

- **Agda**: Only standard filters applied.
- **C** : We include documents based on a keyword whitelist, namely: **"#include <fftw.h>"**, **"#include <fftw3.h>"**, **"#include <rfftw.h>"**, **"#include <gsl"**, **"#include <cblas.h>"**, **"#include <blas.h>"**, **"#include <lapacke.h>"**, **"#include <nlopt.h>"**, **"#include <petsc.h>"**.
- **C++** : We include documents based on a keyword whitelist, namely: **"#include <adept_arrays.h>"**, **"#include <adept.h>"**, **"#include <alglib>"**, **"#include <boost"**, **"#include <armadillo"**, **"#include <blitz"**, **"#include <Eigen"**, **"#include <deal.II"**, **"#include <dlib"**, **"#include <NTL"**, **"#include <mtl"**.
- **Fortran** : Only standard filters applied.
- **GAP** : Only standard filters applied.
- **Haskell** : We filtered the data to only contain files with the following imports: **Numeric.LinearAlgebra**, **Numeric.SpecFunctions**, **Numeric.Vector**, **Statistics**, **Data.Complex**.

- **Idris** : Only standard filters applied.

- **Julia** : We filtered out mislabeled JSON lines files. We removed files larger than 10,000 characters long which both were not files containing tests and which had a lower numerical density than 0.5, and otherwise ignored numerical density. We additionally only accepted files within a specific keyword whitelist, to attempt to control relevance to scientific computing, namely: **"LinearAlgebra"**, **"DifferentialEquations"**, **"Symbolics"**, **"Distributions"**, **"DataFrames"**, **"DynamicalSystems"**, **"Turing"**, **"Gen"**, **"JuMP"**, **"sqrt"**, **"abs"**, **"zeros"**, **"ones"**, **"sin"**, **"cos"**, **"tan"**, **"log"**, **"exp"**, **"integrate"**, **"likelihood"**, **"Matrix"**, $\pi$, **"pi"**, **"rand"**, **"grad"**.

- **Jupyter** : We found that many Jupyter notebook files were large due to containing long cell outputs, such as base64 images, long tracebacks, or other extra JSON cell metadata. We use `nbconvert` to convert notebooks to a markdown format, removing metadata.

- **Maple** : We filtered out files with a size greater than $100,000$ bytes, and found that some files were XML. We filtered all files beginning with an XML declaration.

- **Python** : We filtered notebooks and JSON files out by excluding documents with beginning **"{"** characters, and included only files importing from a fixed list of libraries.

- **R** : We excluded all files beginning with an XML declaration. We additionally filtered out all notebooks, and filtered all files containing MacOS "Resource Fork" files.

- **Tex** : We used a max file size of 10,000,000 bytes. We excluded tex files found in directories named **"latex/"** because these were often auto-generated files, and excluded documents using **gnuplot**. We included only documents containing one of the keywords **"\chapter{"**, **"\chapter*{"**, **"\section{"**, **"\section*{"**, **"\subsection{"**, **"\subsection*{"**, **"\subsubsection{"**, **"\subsubsection*{"**, **"\paragraph{"**, **"\subparagraph{"**, and additionally only included documents identified as English by a classifier from the `langid` package.

For all languages we used within the Stack, unless otherwise stated, we additionally filtered out files with a filesize greater than $1048575$ bytes or with a numerical density (ratio of digit characters to non-digit characters) of $0.5$.

We used v1.2 of the near-deduplicated Stack as a base for processing.

### B.2   PAPERS: ARXIV

We use the entirety of ArXiv, as accessed by Computer (2023) in April 2023. For further information on preprocessing applied to ArXiv, see Computer (2023).

### B.3   WEB: OPENWEBMATH

For the web portion of our training dataset, we use OpenWebMath (Paster et al., 2023).

## C   EVALUATION HARNESS

We implement a variety of math-related tasks and evaluation protocols into a public fork of the Language Model Evaluation Harness (Gao et al., 2021). The Harness provides a model-agnostic framework for standardized, reproducible evaluation of language models.

We add the following tasks for the evaluations in this paper:

- `hendrycks_math_ppl`: Perplexity evaluation on MATH (Hendrycks et al., 2021a) sub-tasks.

- `minif2f_isabelle`: Proof autoformalization in Isabelle on the miniF2F benchmark based on Jiang et al. (2023), with a Portal-to-Isabelle (Jiang et al., 2021) proof checker.

- `minerva_math`: The MATH benchmark with the prompt and Sympy evaluation from Minerva (Lewkowycz et al., 2022).

- `minerva-hendrycksTest`: MMLU-STEM tasks following Lewkowycz et al. (2022).

- `ocw_courses`: The OCW Courses task from Lewkowycz et al. (2022).
- `python_gsm8k`: GSM8k with Python, based on Gao et al. (2022).
- `sympy_math`: MATH with Sympy evaluation.

We include a link to the implementations for these tasks, including full prompts, in our public codebase.

## D  EVALUATION: EXPERIMENT DETAILS

### D.1  ISABELLE INFORMAL-TO-FORMAL THEOREM PROVING

We follow Jiang et al. (2023), allowing the model to issue a call to built-in Isabelle automation in the output proof by generating `sledgehammer`. This calls Sledgehammer (Paulson & Nipkow, 2023) and the list of heuristics listed in Jiang et al. (2023). Following Jiang et al. (2023), as a baseline we use Sledgehammer and the heuristics executed at the beginning of the proof (referred to as Sledgehammer in the main text for brevity). We use a 30-second timeout for Sledgehammer and implement proof checking via Portal-to-Isabelle (Jiang et al., 2021). Refer to the implementation in the Evaluation Harness for further details.

### D.2  LEAN THEOREM PROVING

Theorem proving via tactic prediction involves interacting with a proof assistant after each step of a proof. Implementing these interactions within the evaluation harness is outside the scope of this work. Therefore, for the Lean theorem proving task we use a separate evaluation setup based on an open-source implementation (Welleck, 2023). We include our evaluation code in our public codebase.

**Setup.**  We evaluate on miniF2F (Zheng et al., 2021), which consists of 488 formalized statements from math competitions and undergraduate coursework. Given a formalized statement, the task is to generate a formal proof that is checked by Lean.

We use best first search, commonly used for neural tactic prediction models (e.g., Polu & Sutskever (2020)). Best first search is parameterized by the number of attempts (N), generated tactics per iteration (S), and maximum iterations (T). We define the *search budget* to be the maximum number of generated tactics, $N \times S \times T$. We set our search budget to $N = 1$, $S = 32$, and $T = 100$, less than that of the baseline model. Following Yang et al. (2023), we generate tactics with beam search and use a 10 minute timeout. We adapt the proof search implementation from Welleck (2023), which uses LeanDojo v.1.1.2 (Yang et al., 2023) for interaction. We use Lean 4 miniF2F, using `https://github.com/rah4927/lean-dojo-mew` commit `d00c776260c77de7e70125ef0cd119de6c0ff1de`. Note that the ReProver baseline from (Yang et al., 2023) reports performance with Lean 3.

**Prompt.** We prompt the model with three (state, tactic) examples, shown in Figure 5.

```
"""Given the Lean 4 tactic state, suggest a next tactic.
Here are some examples:

Tactic state:
---
α : Type u_1
r : α → α → Prop†
inst¹ : DecidableEq α†
inst : IsIrrefl α r
⊢ CutExpand r ≤ InvImage (Finsupp.Lex (ᶜr ⊓ fun x x_1 => x ≠ x_1)
    fun x x_1 => x < x_1) ↑toFinsupp
---
Next tactic:
---
rintro s t ⟨u, a, hr, he⟩
---

Tactic state:
---
ι : Type u_1†
I †J : Box ι
x y : ι → ℝ
I J : WithBot (Box ι)
⊢ ↑I = ↑J ↔ I = J
---
Next tactic:
---
simp only [Subset.antisymm_iff, ← le_antisymm_iff,
    withBotCoe_subset_iff]
---

Tactic state:
---
m n : ℕ
h : Nat.coprime m n
⊢ Nat.gcd m n = 1
---
Next tactic:
---
rw [← h.gcd_eq_one]
---

Tactic state:
---
%s
---
Next tactic:
---"""
```

Figure 5: Prompt for the Lean theorem proving experiments.

# E  DATASHEET

We provide a datasheet for Proof-Pile-2, following the framework in Gebru et al. (2021).

| MOTIVATION | |
|---|---|
| **For what purpose was the dataset created?** | Proof-Pile-2 was created for the training or finetuning of domain-specific large language models for general mathematics tasks. |
| **Who created the dataset and on behalf of which entity?** | The dataset was created by the authors of this paper for the purposes of this research project. |
| **Who funded the creation of the dataset?** | The creation of the dataset was funded by the coauthors' grants and employers, as further described in section 5. |
| **Any other comment?** | |
| **COMPOSITION** | |
| **What do the instances that comprise the dataset represent?** | Instances are text-only documents. |
| **How many instances are there in total?** | We detail fine-grained token counts elsewhere in this paper. |
| **Does the dataset contain all possible instances or is it a sample (not necessarily random) of instances from a larger set?** | Our dataset is filtered based on our assessments of quality for the language modeling task. More detail on methodology can be found in Appendix B. |
| **What data does each instance consist of?** | Each instance is a text-only document, alongside metadata about its originating split and filename or location. |
| **Is there a label or target associated with each instance?** | No. |
| **Is any information missing from individual instances?** | Yes, we filter undesired noise, such as base64-encoded images, from some documents. |
| **Are relationships between individual instances made explicit?** | No. |
| **Are there recommended data splits?** | Yes, we release a canonical train, validation, and test split of the dataset, which we follow in this work. |
| **Are there any errors, sources of noise, or redundancies in the dataset?** | We make our best efforts to remove errors or sources of noise, but our dataset will naturally contain documents with errors or noise, and may contain near-duplicate documents. |
| **Is the dataset self-contained, or does it link to or otherwise rely on external resources?** | The dataset is self-contained, but can also be reconstructed based on external publicly available data sources and datasets following our instructions. |
| **Does the dataset contain data that might be considered confidential?** | All documents in Proof-Pile-2 are publicly available online. |

| | |
|---|---|
| **Does the dataset contain data that, if viewed directly, might be offensive, insulting, threatening, or might otherwise cause anxiety?** | We estimate toxic content to be less prevalent in our dataset than other more general web-based datasets, due to its technical focus. However, it is likely to contain such content. |



**COLLECTION**



| | |
|---|---|
| **How was the data associated with each instance acquired?** | Data was largely sourced from existing public subsets, such as the RedPajama dataset (Computer, 2023), OpenWebMath dataset (Paster et al., 2023), and via filtering the Stack (Kocetkov et al., 2022). Some data was collected using the Github API. |
| **What mechanisms or procedures were used to collect the data?** | See above. |
| **If the dataset is a sample from a larger set, what was the sampling strategy?** | We release the entirety of the dataset following the application of our quality filters. We randomly held out validation and test splits from the dataset. |
| **Who was involved in the data collection process and how were they compensated?** | The authors of this paper participated in locating, retrieving, and filtering the dataset. |
| **Over what timeframe was the data collected?** | This data was collected in 2023, with a cutoff date of April 2023 for all subsets with the exception of our Lean proofstep data. |
| **Were any ethical review processes conducted?** | Yes, the authors conducted an informal ethical review internally. |



**PREPROCESSING**



| | |
|---|---|
| **Was any preprocessing/cleaning/labeling of the data done?** | Yes, the authors extensively filtered the dataset subsets in keeping with our expectations for high-quality language modeling data in our domain. See Appendix B for further detail on filtering steps taken. |
| **Was the "raw" data saved in addition to the preprocessed/cleaned/labeled data?** | Raw data can be accessed via reuse of our provided codebase. |
| **Is the software that was used to preprocess/clean/label the data available?** | Yes. We release our codebase, which can be used to reproduce our dataset and its construction process, at `https://github.com/EleutherAI/math-lm`. |



**USES**



| | |
|---|---|
| **Has the dataset been used for any tasks already?** | Yes, this dataset has been used to train the LLEMMA language models as a domain adaptation and continued pretraining corpus. |
| **Is there a repository that links to any or all papers or systems that use the dataset?** | No. |
| **What (other) tasks could the dataset be used for?** | The dataset was specifically targeted as a high quality language modeling corpus for the mathematics domain, but may be useful for general-purpose language modeling or unforeseen other downstream uses. |

| | |
|---|---|
| **Is there anything about the composition of the dataset or the way it was collected and preprocessed/cleaned/labeled that might impact future uses?** | We filtered the dataset with the intent of creating a model useful for mathematical tasks with solely English text. |
| **Are there tasks for which the dataset should not be used?** | The dataset should not be used with the intent to cause harm or for models intended for the purposes of harm. |

### DISTRIBUTION

| | |
|---|---|
| **Will the dataset be distributed to third parties outside of the entity on behalf of which the dataset was created?** | We make the dataset publicly available for reproducibility, analysis, and other further downstream uses. |
| **How will the dataset will be distributed?** | We provide code to replicate the dataset, and release it via the Huggingface Hub. |
| **When will the dataset be distributed?** | The dataset is available immediately. |
| **Will the dataset be distributed under a copyright or other intellectual property (IP) license, and/or under applicable terms of use (ToU)?** | We do not relicense the dataset's components, and do not impose our own use restrictions. |
| **Have any third parties imposed IP-based or other restrictions on the data associated with the instances?** | Not to our knowledge. |
| **Do any export controls or other regulatory restrictions apply to the dataset or to individual instances?** | Not to our knowledge. |

### MAINTENANCE

| | |
|---|---|
| **Who will be supporting/hosting/maintaining the dataset?** | The dataset will be hosted on the HuggingFace Hub and able to be recreated via code at `https://github.com/EleutherAI/math-lm`. The dataset will not be updated post-release. |
| **How can the owner/curator/manager of the dataset be contacted?** | Via email at `za2514@princeton.edu` |
| **Is there an erratum?** | No. |
| **Will the dataset be updated?** | No. |
| **If others want to extend/augment/build on/contribute to the dataset, is there a mechanism for them to do so?** | No. |

Table 10: **Datasheet for** Proof-Pile-2, following the framework introduced by Gebru et al. (2021).

## F    ADDITIONAL RESULTS

### F.1    PROOF AUTOFORMALIZATION

Table 11 shows additional results on Isabelle proof autoformalization, including the union of theorems closed by Sledgehammer and the given language model.

| Method | Autoformalization pass@1 | |
| --- | --- | --- |
| | miniF2F-valid* | miniF2F-test |
| Sledgehammer | 14.72% | 20.49% |
| Code Llama 7b | 16.31% | 17.62% |
| LLEMMA-7b | 20.60% | 22.13% |
| Code Llama 7b ∪ Sledgehammer | 20.17% | 25.00% |
| LLEMMA-7b ∪ Sledgehammer | 25.97% | 27.46% |

Table 11: **Isabelle autoformalization**. *We exclude the 11 examples used in the few-shot prompts. Pass@1 with greedy decoding.

## G    SUPERVISED FINETUNING

A full exploration of finetuning applications for LLEMMA, such as instruction following (Ouyang et al., 2022; Wei et al., 2022), dialogue modeling (Thoppilan et al., 2022; Touvron et al., 2023; Collins et al., 2023), and reward modeling (Cobbe et al., 2021; Lightman et al., 2023) are outside the scope of this work. However, to establish that LLEMMA retains its advantage over other open models when finetuned, we conduct preliminary experiments finetuning LLEMMA-7B on MetaMathQA (Yu et al., 2023), a supervised dataset targeted at the MATH and GSM8k benchmarks. Results are shown in Table 12.

| Initialization | Finetune Dataset | MATH | GSM8k |
| --- | --- | --- | --- |
| Llama 2 7B | WizardMath (Proprietary) | 10.7% | 54.9% |
| Llama 2 7B | MetaMathQA | 19.4% | 66.4% |
| LLEMMA 7B | MetaMathQA | **25.2%** | **66.5%** |
| Llama 2 70B | WizardMath (Proprietary) | 22.7% | 81.6% |
| Llama 2 70B | MetaMathQA | **26.6%** | **82.3%** |

Table 12: Finetuning of various 7B base models on supervised mathematics datasets. All results with a Llama 2 initialization are copied from the literature (Luo et al., 2023; Yu et al., 2023). The LLEMMA 7B finetune is trained with identical hyperparameters to the models in Yu et al. (2023).

## H    QUALITATIVE EXAMPLES

**Dataset overlap.**    Figure 6 shows example false positives when checking $n$-gram overlap with OpenWebMath documents for various $n$. Figure 7 shows an example OpenWebMath document that has 30-gram overlap with a MATH problem, and LLEMMA-7b's generated solution.

**Task outputs.**    Figure 8 shows a generated proof in the informal2formal theorem proving task.

---

**OpenWebMath document**

```
2D affine transformations can be better represented using 2 by 2 matrices, since they
are simply linear combinations of 2 variables. The advantage of this is that the matrices
are associative under multiplication Also, GPUs and modern toolkits are optimised to work
with this representation. As a result, a scale matrix is \begin{bmatrix} s_x & 0 \\ 0 &
s_y \end{bmatrix}, and a rotation matrix is \begin{bmatrix} \cos \theta & -\sin \theta \\
\sin \theta & \cos \theta \end{bmatrix}.

A translation matrix is simply \begin{bmatrix} 1 & \frac{t_x}{y} \\ \frac{t_y}{x} & 1 ...
```

- - - - - - - - - - - - - - - - - - - - - - - - - - - -

**MATH problem**

A rotation centered at the origin takes $\begin{pmatrix} 13 \\ 0 \end{pmatrix}$ to $\begin{pmatrix} 5 \\ -12 \end{pmatrix}$. Which vector does the rotation take $\begin{pmatrix} 0 \\ 1 \end{pmatrix}$ to?

**MATH solution**

The rotation matrix must be of the form $\begin{pmatrix} \cos\theta & -\sin\theta \\ \sin\theta & \cos\theta \end{pmatrix}$. Thus,...

- - - - - - - - - - - - - - - - - - - - - - - - - - - -

**Hit**

```
\cos \theta & -\sin \theta \\ \sin \theta & \cos
```

---

**OpenWebMath document**

```
# Basic Probability

A number is selected at random from 1 through 100, inclusive. What is the probability
that the number is a divisor of 50? Express your answer as a common fraction.

Apr 24, 2019

There are a 100 integers between 1-100, inclusive. Since 50 is $$2*5^2$$, it has
$$(1+1)(1+2)=(2)(3)=6$$ factors. Thus, the answer is
$$\frac{6}{100}=\boxed{\frac{3}{50}}.$$
```

- - - - - - - - - - - - - - - - - - - - - - - - - - - -

**MATH problem**

A number is selected at random from 1 through 100, inclusive. What is the probability that the number is a perfect square?

- - - - - - - - - - - - - - - - - - - - - - - - - - - -

**Hit**

A number is selected at random from 1 through 100, inclusive. What is the probability that the number is a

---

**OpenWebMath document**

```
Fig. 2.

Use values of the most used medicinal plants in the Safi Province (Morocco).

It is also important to note that for the abovementioned medicinal plants, many other
folk uses have been reported in different regions of Morocco. Furthermore,
literature-based proof revealed that these species have proven a wide variety of
biological and pharmacological activities (Table 4, Ref. [14, 17, 19, 20, 21, 23,
24, 26, 28, 30, 31, 34, 35, 36, 38, 39, 65, 66, 67, 68, 69, 70, 71, 72, 73, 74, 75,
76, 77, 78, 79, 80, 81, 82, 83, 84, 85, 86, 87, 88, 89, 90, 91, 92, 93, 94, 95, 96,
97, 98, 99, 100, 101, 102, 103, 104, 105, 106, 107, 108, 109, 110, 111, 112, 113,
114, 115, 116]), which may confirm the different popular applications of ...
```

- - - - - - - - - - - - - - - - - - - - - - - - - - - -

**Generated solution (LLEMMA 7b):**

The first 100 positive integers are 1,2,3,4,5,6,7,8,9,10,11,12,13,14,15,16,17,18,19,20,21,22,23,24,25,26,27,28,29,30,31,32,33,34,35,36,37,38,39,40,41,42,43,44,45,46,47,48,49,50,51,52,53,54,55,56,57,58,59,60,61,62,63,64,65,66,67,68,69,70,71,72,73,74,75,76,77,78,79,80,81,82,83,84,85,86,87,88,89,90,91,92,93,94,95,96,97,98,99,100. We can see that...

Figure 6: **Data overlap:** Example false positives using 10-gram match between MATH solutions and Open-WebMath documents (*top*), 20-gram match between MATH problems and OpenWebMath documents (*middle*), and 30-gram match between LLEMMA-7b's generated solutions and OpenWebMath documents (*bottom*).

**OpenWebMath document**

```
A triangle is formed with edges along the line $y=\frac{2}{3}x+5$,
the $x$-axis, and the line $x=k$. If the area of the triangle is less than $20$,
find the sum of all possible integral values of $k$.

Feb 28, 2018

Look at the graph, here...two triangles are possible :
https://www.desmos.com/calculator/m6wnjpgldq
The  height of the triangles at any point will be formed by

[ (2/3)x + 5 ]

And the bases will be  [ x - (- 7.5)]   = [ x + 7.5]
So....we want to solve this
(1/2) [ (2/3)x + 5 ] [ x + 7.5 ]  =  20
[ (2/3)x + 5 ] [ x + 7.5]  =  40
(2/3)x^2 + 5x + 5x  + 37.5 = 0
(2/3)x^2 + 10x - 2.5  =  0

Using a little technology.....the max x  value for the
triangle formed above the x axis will be = .246

And the min x value for the triangle formed below the x axis will be = -15.246

With the given boundaries, the  integer sums  of all possible x values of k giving
triangles with an area < 20 units^2  =

[ (-15) + (-14 ) + (-13) + ...+ ( -2) + ( - 1 ) + 0 ]  =
-  (15) (16) / 2   =
-120

Feb 28, 2018
```

---

**MATH Problem:** A triangle is formed with edges along the line $y = \frac{2}{3}x + 5$, the $x$-axis, and the line $x = k$. If the area of the triangle is less than 20, find the sum of all possible integral values of $k$.
**Answer:** -120

---

**Generated solution (LLEMMA 7b):**
The triangle has vertices $(k, 0)$, $(0, 5)$, and $(k, \frac{2}{3}k + 5)$. The base of the triangle has length $k$, and the height of the triangle has length $\frac{2}{3}k + 5$. The area of the triangle is therefore

$$A = \frac{1}{2}bh = \frac{1}{2}k(\frac{2}{3}k + 5) = \frac{1}{3}k^2 + \frac{5}{2}k.$$

Since the area of the triangle is less than 20, we have

$$\frac{1}{3}k^2 + \frac{5}{2}k < 20 \Rightarrow \frac{1}{6}k^2 + \frac{5}{4}k - 10 < 0.$$

This inequality is true if and only if $\frac{1}{6}k^2 + \frac{5}{4}k - 10$ is positive. Since $\frac{1}{6}k^2 + \frac{5}{4}k - 10$ is a quadratic with positive leading coefficient, it is positive if and only if its vertex is less than 0. The vertex of $\frac{1}{6}k^2 + \frac{5}{4}k - 10$ is $\left(-\frac{10}{3}, -\frac{125}{18}\right)$. Since $-\frac{10}{3} < 0$, we have that $\frac{1}{6}k^2 + \frac{5}{4}k - 10 > 0$ if and only if $k < -\frac{10}{3}$. Therefore, the only possible integral values of $k$ are $k = 0, 1, 2, 3, 4$. The sum of these values is $\boxed{10}$. Final Answer: The final answer is 10.

---

Figure 7: **Data overlap:** Example OpenWebMath document that has a 30-gram overlap with the given MATH problem, and LLEMMA-7b's generated solution.

**Problem:** If $3a + b + c = -3, a + 3b + c = 9, a + b + 3c = 19$, then find $abc$. Show that it is -56.

---

**Informal Proof (Human-written):** Summing all three equations yields that $5a + 5b + 5c = -3 + 9 + 19 = 25$. Thus, $a + b + c = 5$. Subtracting this from each of the given equations, we obtain that $2a = -8, 2b = 4, 2c = 14$. Thus, $a = -4, b = 2, c = 7$, and their product is $abc = -4 \times 2 \times 7 = -56$.

---

**Formal Statement and Proof:**

```
theorem mathd_algebra_338:
  fixes a b c :: real
  assumes "3 * a + b + c = -3" and "a + 3 * b + c = 9" and "a + b + 3 * c = 19"
  shows "a * b * c = -56"
proof -
    (* Summing all three equations yields that 5a + 5b + 5c = -3 + 9 + 19 = 25.
    Thus, a + b + c = 5. *)
    have "5 * a + 5 * b + 5 * c = -3 + 9 + 19" using assms <ATP>
    then have "5 * (a + b + c) = 25" <ATP>
    then have "a + b + c = 5" <ATP>
    (* Subtracting this from each of the given equations, we obtain that  2a =
    -8, 2b = 4, 2c = 14. Thus, a = -4, b = 2, c =7, and their product is  abc =
    -4 \times 2 \times 7 = -56. *)
    then have "2 * a = -8" "2 * b = 4" "2 * c = 14" using assms <ATP>
    then have "a = -4" "b = 2" "c = 7" <ATP>
    then show ?thesis <ATP>
qed
```

Figure 8: **Informal-to-formal proving**. The model is given the problem, informal proof, and formal statement, following Jiang et al. (2023). It generates a formal proof (starting with `proof -`) containing Isabelle code, comments (`(*...*)`) that align the informal and formal proofs, and calls to an automated prover (shown as *<ATP>*). The proof is from LLEMMA-7b with greedy decoding.

