# OpenReview forum: "Llemma: An Open Language Model for Mathematics"
_ICLR.cc/2024/Conference — ICLR 2024 poster_

### Official Review · Reviewer_Ch2c · 2023-10-18

**Soundness:** 3 good
**Presentation:** 3 good
**Contribution:** 3 good
**Rating:** 8
**Confidence:** 3

**Summary:**

The paper proposed continual training with math/code data and achieved very competitive results on the related tasks. The evaluation is comprehensive and ablation studies are sound.

**Strengths:**

1. This paper showcases that if we continue pretraining an LLM on a specific domain, we are able to get good performance. The authors established a good way to do domain adaptation.
2. Data and training code are open-sourced, expect to have high reproducibility
3. Ablation study is comprehensive.
4. Writing is well-organized

**Weaknesses:**

- The authors are comparing their results with `Minerva`, however since the datasets and its mixture, model architecture, training methods are different, we don't know which part contributes to the good performance.
- We don't know if there training from scratch or starting from other LLM (like llama2 base) could be as impactful as well

**Questions:**

- One thing would be interesting to know is that if pretrained from scratch, will it be better or not?
- Do we still need fine-tuning in this case

---

> ### Author Response · Authors · 2023-11-16
>
> We thank the reviewer for indicating that we have “established a good way to do domain adaptation”, that our “ablation study is comprehensive”, and that the paper is well-organized. We hope to address their questions and criticisms below.
>
> **The authors are comparing their results with Minerva, however since the datasets and its mixture, model architecture, training methods are different, we don't know which part contributes to the good performance.**
>
> We use Minerva as a baseline because it is a state-of-the-art methodology for adapting language models to mathematics. We would have happily performed a detailed ablation study comparing each component of Minerva’s data and training pipeline with Llemma’s. However, because Minerva is entirely closed source, such a study is impossible.
>
> We would like to note that, wherever possible, in our comparisons between the two models we evaluate Llemma and Minerva on precisely the same evaluation setup, such as the use of the exact few-shot prompt and answer extraction used by Minerva, and as such our downstream comparisons are as detailed as Minerva’s closed-source nature allows.
>
> **We don't know if training from scratch or starting from another LLM (like llama2 base) could be as impactful as well. One thing would be interesting to know is that if pretrained from scratch, will it be better or not?**
>
> We certainly would have been interested in a head-to-head comparison of a random initialization, a Llama-2 initialization, and a Code Llama initialization for training Llemma. However, the compute cost of such experiments is prohibitive. At current prices of an AWS 8xA100 instance, training Llemma 7B costs around 95,000 USD and training Llemma 34B costs around 195,000 USD.
>
> We chose to initialize Llemma with Code Llama weights instead of Llama 2 weights for several reasons. One is that coding and theorem proving are important use cases for Llemma, and Llama 2 has weak coding capabilities. Furthermore, code pretraining appears to boost more general reasoning capabilities, as evidenced by Code Llama’s superior MATH score compared to Llama 2.
>
> **Do we still need fine-tuning?**
>
> Yes, finetuning Llemma on downstream applications is beneficial. For example, in appendix G, we demonstrate that finetuning Llemma on supervised datasets targeted at mathematical problem solving boosts its accuracy on MATH and GSM8k. Furthermore, finetuning can align Llemma’s outputs to the desired style or application, such as boosting the chat dialog capabilities of the model.

---

### Official Review · Reviewer_JQiP · 2023-10-31

**Soundness:** 2 fair
**Presentation:** 3 good
**Contribution:** 3 good
**Rating:** 6
**Confidence:** 2

**Summary:**

In this submission, the authors released a new large-scale dataset for training mathematically-specified
large language models. Additionally, based on the dataset, a new base model called LLemma is trained and released, which outperforms existing models like Code LLaMA and Minerva. Moreover, the dataset and the baseline are openly released, which helps to boost the research in AI4Math.

**Strengths:**

1. The paper is well-written and easy to follow. The details of data collection and training instructions are provided. Moreover, both data and model are released.

2. The experimental part is solid. I especially like the ablation study of data mixture components given in Table 4, which helps to reveal the contribution of different data sources.

**Weaknesses:**

1. Technically, this submission is not so interesting, and I cannot learn anything new in the aspect of methodology. The claims and experimental results are natural — fine-tuning Code Llama on a larger mathematically specific dataset helps to improve its performance. Note that this is a very personal opinion:) It is OK if this work targets the topic of datasets and benchmarks.
2. I hope that the authors can discuss more about their future plans in the section of Conclusion. In the field of AI4Math, the performance of the current LLM is still limited. What will the authors do in the next step? Will they continuously enlarge the dataset and/or model? Is this the final solution to AI4Math?
3. The comparison between Code Llama and the proposed Llemma is not fair. It demonstrates the usefulness of the Proof-Pile-2 dataset. It would be nice if the authors could finetune Llama2 directly based on the Proof-Pile-2 dataset and show the model performance.

**Questions:**

Why did the authors train the 7B model for 200B tokens and the 34B (the larger) model for 50B (the fewer) tokens?

---

> ### Author Response · Authors · 2023-11-16
>
> We thank the reviewer for their thoughtful comments. Below, we address each of the concerns raised point-by-point.
>
> **I cannot learn anything new in the aspect of methodology**
>
> Llemma provides a full recipe for adapting a language model to mathematics through continued pretraining. By full recipe, we mean that a lab could use our paper, code, data, and models to understand how we processed the data, picked a data mixture (section 3.4), trained the model, and evaluated it (including new methodologies for studying data overlap, and for evaluating language models few-shot for formal theorem proving).
>
> At the scale we considered, a full recipe of this form is increasingly rare. As a result, we hope that our work gives insight into the methodology of training language models for a specialized domain. In the words of Reviewer Ch2c, Llemma “established a good way to do domain adaptation”.
>
> **I hope that the authors can discuss more about their future plans**
>
> We believe a strong mathematics base model is a useful (if not necessary) component of several outgoing research directions. We mentioned these in the introduction and conclusion, such as reinforcement learning and reward modeling for reasoning, investigating the limits of domain-specific models, and gaining a better understanding of algorithmic generalization. Llemma is also unique in its coverage of formal mathematics, which we see as a fruitful area for the future.
>
> **The comparison between Code Llama and the proposed Llemma is not fair. It demonstrates the usefulness of the Proof-Pile-2 dataset. It would be nice if the authors could finetune Llama2 directly based on the Proof-Pile-2 dataset and show the model performance.**
>
> We would like to clarify that the purpose of finetuning Code Llama on the Proof-Pile-2 to yield Llemma is to demonstrate the usefulness of the Proof-Pile-2. That is, the core claim of our paper is that training on unstructured mathematical text can greatly increase the quantitative reasoning performance of a pre-trained language model.
>
> That said, it still would have been interesting to train a version of Llemma initialized from Llama 2 to understand how much better Llemma is than Code Llama at mathematics when controlled for training compute. However, the computational cost of such an experiment is prohibitive. At the current AWS rate of 32.70 USD/hour for an 8xA100 instance, it would cost 95,000 USD to retrain Llemma 7B and 195,000 USD to retrain Llemma 34B.
>
> **Why did the authors train the 7B model for 200B tokens and the 34B (the larger) model for 50B (the fewer) tokens?**
>
> The decision to train the 34B model for fewer tokens was forced on us by our compute budget. We simply did not have more resources than the total 70,000 GPU hours we used to train Llemma 7B for 200B tokens and Llemma 34B for 50B tokens.
>
> We hope the above clarifications address the  reviewers' concerns around our experimental design, and if so, wonder whether they may be willing to raise our submission’s score.

---

### Official Review · Reviewer_HYfd · 2023-11-07

**Soundness:** 3 good
**Presentation:** 3 good
**Contribution:** 3 good
**Rating:** 6
**Confidence:** 4

**Summary:**

This paper present a new model, namely LLEMMA, an open language model for mathematics. The model is used to solve math-related problems such as proving or calculators. The major contribution of the paper is opensourcing the model, as well as the training data and codes. The evaluation results show that the performance is encouraging.

**Strengths:**

1. The model and code are opensourcing.
2. It has proposed the Proof-Pile-2 data and will be released as well.
3. It has shown a training method that works well by fine-tuning a llama2 based pretrained model.

**Weaknesses:**

1. It doesn't have much ablation study or analysis about the data configuration. For example, why we should use which part of data.
2. It doesn't have much novelties, other than a model that experimentally looks ok.

**Questions:**

1. Have you experimented with different combination of training data?
2. How did you perform instruction fine-tuning?

---

> ### Author Response · Authors · 2023-11-16
>
> We thank the reviewer for their thoughtful remarks. Below, we address each of their concerns about our submission point-by-point.
>
> **It doesn't have much ablation study or analysis about the data configuration. For example, why we should use which part of data. Have you experimented with different combinations of training data?**
>
> In section 3.4 and Table 4 of our paper, we report an ablation study we carried out to determine the optimal mixture of data sources. Note that many prior works on language modeling [5, 6] provide little justification for their data mixture. We provide important confirmation that large-scale language modeling performance is sensitive to the precise choice of data mixture.
>
> **It doesn't have much novelties, other than a model that experimentally looks ok.**
>
> We would like to point out a number of aspects of our work that are novel:
> 1. Few-shot formal theorem proving: Prior work using language models for formal theorem proving has either relied on models finetuned for a specific theorem proving system [1, 2], or proprietary models finetuned on private datasets [2]. We demonstrate the first instance of an open base model achieving strong theorem proving capabilities without fine-tuning. One reason this is a useful capability is that supervised finetuning data for formal theorem proving and autoformalization are scarce, which means being able to finetune from a strong base model is critical.
> 2. Memorization study: in our memorization study (section 3.5), we find that our model does not attain a higher accuracy on problems that appear in the training set compared to those that do not. This is a counterintuitive finding that warrants awareness among the research community and further investigation.
>
> Furthermore, we view Llemma not just as the outcome of research, but also as research infrastructure that can support future work. For example, many works on machine learning theorem proving [1, 2, 3] may benefit greatly from a stronger mathematics language model.
>
> **How did you perform instruction fine-tuning?**
>
> All results in the main body of the paper show Llemma’s performance via few-shot prompting, without applying any finetuning.
>
> In Appendix G and Table 11 of our paper, we also demonstrate that Llemma is able to substantially improve its performance on MATH and GSM8k when fine-tuned on MetaMathQA, a supervised finetuning dataset targeted at mathematical problem solving. For these experiments, we use exactly the same methodology as [4]. Finetuning code will be released in our open-source repository upon acceptance.
>
> We hope that after reading our clarifications, the reviewer is willing to raise their score.
>
> **References**
>
> [1] Kaiyu Yang, Aidan M. Swope, Alex Gu, Rahul Chalamala, Peiyang Song, Shixing Yu, Saad Godil, Ryan Prenger, & Anima Anandkumar. (2023). LeanDojo: Theorem Proving with Retrieval-Augmented Language Models.
>
> [2] Emily First, Markus N. Rabe, Talia Ringer, & Yuriy Brun. (2023). Baldur: Whole-Proof Generation and Repair with Large Language Models.
>
> [3] Amitayush Thakur, Yeming Wen, & Swarat Chaudhuri. (2023). A Language-Agent Approach to Formal Theorem-Proving.
>
> [4] Longhui Yu, Weisen Jiang, Han Shi, Jincheng Yu, Zhengying Liu, Yu Zhang, James T. Kwok, Zhenguo Li, Adrian Weller, & Weiyang Liu. (2023). MetaMath: Bootstrap Your Own Mathematical Questions for Large Language Models.
>
> [5] Leo Gao, Stella Biderman, Sid Black, Laurence Golding, Travis Hoppe, Charles Foster, Jason Phang, Horace He, Anish Thite, Noa Nabeshima, Shawn Presser, & Connor Leahy. (2020). The Pile: An 800GB Dataset of Diverse Text for Language Modeling.
>
> [6] Hugo Touvron, Thibaut Lavril, Gautier Izacard, Xavier Martinet, Marie-Anne Lachaux, Timothée Lacroix, Baptiste Rozière, Naman Goyal, Eric Hambro, Faisal Azhar, Aurelien Rodriguez, Armand Joulin, Edouard Grave, & Guillaume Lample. (2023). LLaMA: Open and Efficient Foundation Language Models.

---

### Author Response · Authors · 2023-11-16

We thank the reviewers for their thoughtful commentary. We are pleased to hear the reviewers commend our open-sourcing of all project artifacts, the strong evaluation scores of our model, and the clarity of our exposition.

We have posted a rebuttal to each review that addresses the reviewer’s comments point-by-point. Furthermore, we will use the remainder of this paragraph to remark on themes that recurred throughout the reviews. One common question was why we did not perform additional ablation studies, such as training versions of Llemma from scratch or from a Llama-2 initialization. We would have loved to perform such experiments, but their computational cost is prohibitive. At the current AWS on-demand price of 32.70 USD/hour for an 8xA100 instance, it costs approximately 95,000 USD to retrain Llemma 7B and 195,000 USD to retrain Llemma 34B. Additionally, although all reviewers were in favor of accepting the paper, some remarked on limited novelty as a drawback of our work. However, we would like to point out that some of our experimental findings, such as few-shot theorem proving and data-contamination not affecting accuracy, are novel and important. Furthermore, Llemma serves as research infrastructure that can enable future work on AI for mathematics.

We have made one minor revision to our submission. At the time of the submission deadline, the MetaMathQA authors had not released the full version of their dataset. Since then, the authors have released their full dataset, and we have updated appendix G to include results that finetune on the full dataset.

---

### Meta-Review · Area_Chair_JjYe · 2023-12-13

**Metareview:**

This paper introduces LLEMMA, a new open language model specifically designed for mathematics. It is good at addressing tsks like proving theorems or performing calculations. The major contribution of this work lies in continuing pre-training the model with existing datasets, along with its open source effort. The evaluation of LLEMMA demonstrates promising performance.

The authors argue that "the core claim of our paper is that training on unstructured mathematical text can greatly increase the quantitative reasoning performance of a pre-trained language model." Both the reviewers and I mostly agree with the argument. However, as reviewer JQiP notes that this effort --- focusing on enhancing a specific aspect of large language models (LLMs) by incorporating new datasets into an existing model without preserving its general capabilities --- does not generate new insights into LLMs.

Despite this, the work is well-executed and yields interesting results, receiving positive reviews (8, 6, 6). Taking everything into account, the paper is recommended for acceptance.

**Justification For Why Not Higher Score:**

The work is well-executed and yields interesting results. However, it does not offer insights into our understanding of training a strong and general LLM and the LLM community is generally familiar with works of the similar type. Therefore, a poster presentation is recommended.

**Justification For Why Not Lower Score:**

n/a

---

### Decision · Program_Chairs · 2024-01-16

Accept (poster)